# The Detrimental Clinical Associations of Anxiety and Depression with Difficult Asthma Outcomes

**DOI:** 10.3390/jpm12050686

**Published:** 2022-04-26

**Authors:** Wei Chern Gavin Fong, Ishmail Rafiq, Matthew Harvey, Sabina Stanescu, Ben Ainsworth, Judit Varkonyi-Sepp, Heena Mistry, Mohammed Aref Kyyaly, Clair Barber, Anna Freeman, Tom Wilkinson, Ratko Djukanovic, Paddy Dennison, Hans Michael Haitchi, Ramesh J. Kurukulaaratchy

**Affiliations:** 1School of Clinical and Experimental Sciences, University of Southampton, Southampton SO16 6YD, UK; wcf1g18@soton.ac.uk (W.C.G.F.); j.varkonyi-sepp@soton.ac.uk (J.V.-S.); hmistry@lji.org (H.M.); aref.kyyaly@solent.ac.uk (M.A.K.); a.freeman@soton.ac.uk (A.F.); t.wilkinson@soton.ac.uk (T.W.); r.djukanovic@soton.ac.uk (R.D.); h.m.haitchi@soton.ac.uk (H.M.H.); 2David Hide Asthma and Allergy Research Centre, Isle of Wight NHS Trust, Isle of Wight PO30 5TG, UK; 3Faculty of Medicine, University of Southampton, Southampton SO16 6YD, UK; ir2g18@soton.ac.uk; 4NIHR Biomedical Research Centre, University Hospitals Southampton NHS Foundation Trust, Southampton SO16 6YD, UK; matthew.harvey@uhs.nhs.uk (M.H.); ba548@bath.ac.uk (B.A.); c.barber@soton.ac.uk (C.B.); paddy.dennison@uhs.nhs.uk (P.D.); 5Department of Psychology, University of Southampton, Southampton SO17 1BJ, UK; s.stanescu@soton.ac.uk; 6Department of Psychology, University of Bath, Bath BA2 7AY, UK; 7Department of Respiratory Medicine, University Hospitals Southampton NHS Foundation Trust, Southampton SO16 6YD, UK

**Keywords:** asthma, mental health, cohort studies, mood disorders, anxiety disorders

## Abstract

Difficult asthma describes asthma in which comorbidities, inadequate treatment, suboptimal inhaler technique and/or poor adherence impede good asthma control. The association of anxiety and depression with difficult asthma outcomes (exacerbations, hospital admissions, asthma control, etc.) is unclear. This study assessed the clinical associations of anxiety and depression with difficult asthma outcomes in patients with a specialist diagnosis of difficult asthma. Using real-world data, we retrospectively phenotyped patients from the Wessex Asthma Cohort of Difficult Asthma (*N* = 441) using clinical diagnoses of anxiety and depression against those without anxiety or depression (controls). Additionally, we stratified patients by severity of psychological distress using the Hospital Anxiety and Depression Scale (HADS). We found that depression and/or anxiety were reported in 43.1% of subjects and were associated with worse disease-related questionnaire scores. Each psychological comorbidity group showed differential associations with difficult asthma outcomes. Anxiety alone (7.9%) was associated with dysfunctional breathing and more hospitalisations [anxiety, median (IQR): 0 (2) vs. controls: 0 (0)], while depression alone (11.6%) was associated with obesity and obstructive sleep apnoea. The dual anxiety and depression group (23.6%) displayed multimorbidity, worse asthma outcomes, female predominance and earlier asthma onset. Worse HADS-A scores in patients with anxiety were associated with worse subjective outcomes (questionnaire scores), while worse HADS-D scores in patients with depression were associated with worse objective (ICU admissions and maintenance oral corticosteroid requirements) and subjective outcomes. In conclusion, anxiety and depression are common in difficult asthma but exert differential detrimental effects. Difficult asthma patients with dual anxiety and depression experience worse asthma outcomes alongside worse measures of psychological distress. There is a severity-gradient association of HADS scores with worse difficult asthma outcomes. Collectively, our findings highlight the need for holistic, multidisciplinary approaches that promote early identification and management of anxiety and depression in difficult asthma patients.

## 1. Introduction

Asthma is one of the most prevalent chronic respiratory diseases worldwide and is associated with substantial morbidity, ranking 16th among leading causes of years lived with disability [1]. Among those with asthma, a large proportion suffer from, and are at risk of, comorbid psychological states, namely depression and anxiety [2,3]. The pathobiological mechanisms by which these conditions might link with asthma remain poorly understood. Nevertheless, there are accumulating data describing the evidence for associations of depression and anxiety with poor outcomes in asthma, including worse quality of life, worse asthma control, asthma exacerbations and related hospitalisations [4,5,6,7,8,9]. However, there is an important gap in the understanding of the specific impact of these conditions among patients with difficult-to-treat (difficult) asthma that is defined by higher asthma symptom burden and treatment need. Difficult asthma is best characterised as a multimorbid state imposing a significant burden at individual, health, economic and societal levels [6,10,11]. It is known that anxiety and depression interact with other comorbidities in asthma [10], but the nature of such interactions specifically within difficult asthma remain unclear [12]. This is particularly relevant now, given a likely rise in mental health problems arising from the COVID-19 pandemic [13], which has disproportionately impacted patients with difficult asthma. It is therefore imperative for clinicians to develop a clearer understanding of the complex interplay between anxiety and depression and the complex entity of difficult asthma.

This study sought to fill this gap in our knowledge by characterising the association between anxiety and/or depression and difficult asthma outcomes using the enrolment data of the well-characterised, longitudinal Wessex Asthma Cohort of Difficult Asthma (WATCH) [14]. We hypothesised that those with anxiety and/or depression would have worse difficult asthma outcomes and that this association would be influenced by the severity of psychological distress.

## 2. Materials and Methods

This was a cross-sectional analysis of the baseline data of the WATCH cohort. WATCH is a prospective observational study of patients with difficult asthma based at the University Hospital Southampton National Health Service Foundation Trust, Southampton, United Kingdom, which serves the population from a wider region in southern England. All patients met the British Thoracic Society “high dose therapies” and/or “continuous or frequent use of oral corticosteroids (OCS)” steps [15] which correspond to Step 4 and Step 5 of the Global Initiative for Asthma (GINA) guidelines [16]. The data reflect the extensive clinical characterisation consistent with a tertiary specialist centre for difficult asthma. This characterisation was conducted on all patients at enrolment and included detailed clinical-, health- and disease-related questionnaires such as ACQ6 (Asthma Control Questionnaire 6), the Nijmegen Questionnaire (to assess dysfunctional breathing status), SGRQ (St. George’s Respiratory Questionnaire, to assess associated health impairment on a 0–100 scale) and HADS (Hospital Anxiety and Depression Scale) [14]. For all of these questionnaires, high scores imply worse status (respective cut-offs for abnormality being ACQ6 > 1.5, Nijmegen score > 23 and HADS > 7). Objective measures included anthropometry, the allergy skin prick test (SPT), FeNO (fractional exhaled nitric oxide) and spirometry (forced expiratory volume in 1 s [FEV_1_], forced vital capacity [FVC], FEV_1_/FVC and mid-expiratory flow [MEF_25–75_]). The detailed study methodology is described in [14]. This study was approved by the West Midlands Solihull Research Ethics Committee (REC reference: 14/WM/1226) and written, informed consent was obtained from all participants.

The diagnoses of anxiety and depression represented historical physician-based clinical diagnoses in patient medical records, which were mostly made by primary care clinicians. These diagnoses were obtained by retrospectively reviewing medical records in the period before the point of the WATCH enrolment. Diagnostic definitions of other comorbidities are in Appendix A.

### Analysis

Statistical analyses were performed with SPSS 27 (IBM Corp., Amonk, NY, USA). To address our aims, the following analyses were performed:We used enrolment data to characterise WATCH patients with anxiety alone against patients with no clinical diagnosis of anxiety or depression (controls). This was repeated for depression alone and for dual anxiety and depression;In patients with the associated psychological comorbidity, the HADS-A (anxiety) and -D (depression) scores were utilised to stratify and characterise patients based on their self-reported severity of psychological distress [17]. The HADS cut-offs used were normal (0–7), mild (8–10), moderate (11–14) and severe (≥15) [17];To further elucidate the association of psychological distress severity with difficult asthma outcomes, we compared patients at polar opposites, i.e., those with ‘severe’ anxiety and/or depression (defined as a clinical diagnosis of anxiety and/or depression and ‘severe’ HADS-A or HADS-D scores) against those who had no anxiety and/or depression (defined as no anxiety, no depression and ‘normal’ HADS-A and HADS-D scores).

We did not undertake multiple comparisons with associated post hoc adjustment as our analysis plan was determined a priori [18] rather than randomly selected.

Continuous variables were presented as mean (standard deviation [SD]) or median (interquartile range [IQR]). Categorical data were presented as proportions (frequency). Data were analysed using independent *t*-tests, the Mann—Whitney U test, the chi-squared test, Fisher’s exact test, one-way ANOVA or Kruskal—Wallis test, as appropriate. Missing data were handled using pairwise deletion in these analyses. Multivariate logistic regression (backward stepwise variable selection) was performed to assess factors independently associated with the dual anxiety and depression and the ‘severe psychological comorbidity’ groups, respectively, using variables that had a *p*-value of <0.1 in univariate analysis. Missing data were handled using listwise deletion for logistic regression procedures, and *p*-values of <0.05 were deemed statistically significant.

## 3. Results

Data were available on psychological comorbidity status in 441 of the 500 WATCH patients at study enrolment. The overall cohort was predominantly female (65.3%), with a median age of 54.0 years, a median age of asthma onset of 19.0 years and with a median BMI of 29.7. Of these, 35 (7.9%) had clinically diagnosed anxiety (without overlapping depression), 51 (11.6%) had depression (without overlapping anxiety) and 104 (23.6%) had dual anxiety and depression. A control group of 251 (56.9%) patients had neither diagnosis.

### 3.1. Anxiety Alone in Difficult Asthma

Dysfunctional breathing was the only comorbidity significantly more prevalent in patients with anxiety alone when compared to controls (Table 1). ACQ6, HADS, SGRQ (activity and impact) and Nijmegen scores were all significantly poorer in those with anxiety alone (Table 2). Additionally, those with anxiety alone had experienced significantly more hospitalisations in the previous 12 months (Table 3). There were no other notable differences with respect to age of asthma onset, sex, current age, BMI, smoking status, other comorbidities, aeroallergen sensitivity or other asthma outcome measures. Enrolment blood eosinophil count, total immunoglobulin E (IgE), FeNO and spirometry also did not differ significantly (Appendix A).

### 3.2. Depression Alone in Difficult Asthma

When compared to controls, obstructive sleep apnoea (OSA), obesity (BMI ≥ 30) and smoking (Table 1, Table 2 and Table 3) were significantly more prevalent in patients with depression alone, while nasal polyposis was significantly less common in those with depression alone. BMI, ACQ6, HADS-A and -D, SGRQ (activity and impact) and Nijmegen scores were also all significantly higher in those with depression alone (Table 1 and Table 2).

Several %PRED spirometry measures were significantly worse in those with depression alone than in the control group (Appendix A). These included Pre-BD (bronchodilator) FEV_1_ [Depression: 54.6%, Control: 74.9%, *p* < 0.005], Pre-BD FVC [Depression: 74.6%, Control: 89.5%, *p* < 0.05], Pre-BD MEF_25–75_ [Depression: 23.7%, Control: 43.6%, *p* < 0.005] and Clinic (post-BD) FVC [Depression Post-BD FVC: 84.1, Control Post-BD FVC: 91.5, *p* < 0.05]. FeNO was also significantly lower in patients with depression alone (12.7 ppb vs. 23.5 ppb, *p* < 0.05; Appendix A). There were no other notable differences, including age-of-onset, sex, current age, bronchiectasis ever, rhinitis ever, atopic status by skin prick test, the presence of other comorbidities, other asthma outcome measures, enrolment blood eosinophil counts or total immunoglobulin E counts.

### 3.3. Dual Anxiety and Depression in Difficult Asthma

When compared to controls, there were significantly more females amongst patients with dual anxiety and depression. Additionally, these patients were significantly younger, exhibited higher BMIs and exhibited an earlier asthma onset (Table 1). Obesity, other psychological comorbidities (not anxiety or depression, e.g., post-traumatic stress disorder, obsessive—compulsive disorder, personality disorders, schizophrenia), dysfunctional breathing, OSA and smoking were also significantly more common in dual-disease patients (Table 1). However, nasal polyposis was significantly less common in this group.

Furthermore, ACQ6, HADS-A and -D, Nijmegen and all SGRQ domains were significantly worse in the dual group than in controls (Table 2). Spirometry did not differ statistically, but FeNO [Dual Disease: 18.9 ppb, Control: 23.5 ppb, *p* < 0.05] was significantly lower in the dual-disease group (Appendix A). Asthma exacerbations (requiring OCS burst or increase in maintenance OCS [mOCS]) and asthma hospitalisations in the previous 12 months were also significantly higher in the dual-disease group (Table 3). There were no other notable differences, including enrolment eosinophil/total immunoglobulin E counts, atopic status by skin prick test, bronchiectasis—ever, the presence of other comorbidities or other asthma outcome measures.

In multivariate analyses, female sex, younger age, a higher prevalence of ever smoking, OSA, longer asthma duration at enrolment and other psychiatric comorbidities were independently associated with dual anxiety and depression (Table 4).

Summary clinical and phenotypic features of patients with clinically diagnosed anxiety, depression and dual anxiety and depression are shown in Figure 1.

### 3.4. Distribution of HADS-A and -D Strata

Of participants with HADS-A and HADS-D data available, Figure 2a,b illustrate the distribution of HADS-A and HADS-D strata (normal, mild, moderate and severe) across the different groups. The dual anxiety and depression group constituted the largest proportion of patients in the ‘severe’ HADS-A (68.8%) and HADS-D (60.0%) strata. Conversely, the control group constituted the largest proportion of patients in the ‘normal’ HADS-A (72.2%) and HADS-D (67.2%) strata. However, more than 50% of patients in each psychological status group fell within the ‘normal’ HADS-D strata. Furthermore, 9.4% (20/213) and 5.1% (11/214) of controls fell into the moderate-to-severe HADS-A and moderate-to-severe HADS-D strata, respectively.

### 3.5. Clinical Outcomes by Stratification with HADS Scores

Among those with depression, there were significant differences across the four HADS-D strata with regard to ACQ6, the SGRQ domains, ICU admissions and mOCS (Table 5). Those in the severe HADS-D strata had the highest prevalence of mOCS dependency (81.8%), worst ACQ6 (median: 4.7) and worst SGRQ domain scores. Additionally, those in the moderate HADS-D strata had the highest prevalence of asthma ICU admissions, ever (60%). There were no significant differences in terms of spirometry, FeNO or other outcome measures (Table 5, Appendix A).

Among dual anxiety and depression patients (Table 6), there were also significant differences across the four HADS-D strata with regard to ACQ6, SGRQ scores, asthma exacerbations, asthma ICU admissions, intubation for acute asthma and mOCS status. There were also significant differences in clinic (post-BD) FVC [HADS-D normal = 91.72; HADS-D mild = 91.94; HADS-D moderate = 92.35; HADS-A severe = 69.18, *p* = 0.020], (Appendix A). FeNO and other lung-function measures did not differ significantly. Those in the moderate and severe HADS-D strata had the most exacerbations (median = 5) and the highest prevalence of ICU admissions, ever (57.1%), and intubations (28.6%) for asthma, ever. Additionally, those in the severe strata exhibited the worst ACQ6 (median = 4.5), SGRQ scores, poorest post-BD FVC and the highest prevalence of mOCS dependency (87.5%).

Among those with anxiety (Appendix A) and those with dual anxiety and depression (Appendix A), the four HADS-A strata were only significantly different with regard to ACQ6 and SGRQ scores. In both of these patient groups, those within the normal HADS-A strata had the best ACQ6 and SGRQ scores. Additionally, in those with dual anxiety and depression, SGRQ Activity scores were not significantly different across the HADS-A strata.

### 3.6. Severe Psychological Comorbidity Is Associated with Worse Asthma Outcomes

Compared to patients with no anxiety or depression and normal HADS-A and -D scores (Appendix A), those with ‘severe anxiety and/or depression’, exhibited significantly worse ACQ6, lost more days from work, exhibited higher Nijmegen scores, worse SGRQ scores and worse FeNO. Additionally, those with severe disease were younger, exhibited a greater prevalence of obesity, higher BMIs, more OSA and more dysfunctional breathing. There were no other statistically significant differences.

In multivariate analysis, female sex and higher ACQ6 were independently associated with ‘severe anxiety and/or depression’.

## 4. Discussion

Our study provides novel real-world insight into the clinical associations of anxiety and depression with difficult asthma outcomes. Our key finding is that anxiety and depression are associated with other physical comorbidities common in difficult asthma and are also associated with poor difficult asthma outcomes. We also uniquely showed the multifaceted yet differential associations of anxiety and depression with difficult asthma. Furthermore, to our knowledge, we are the first to stratify the impact of clinically diagnosed anxiety and depression on difficult asthma outcomes using HADS-A and -D scores, finding that greater psychological distress was associated with worse difficult asthma outcomes. Additionally, the co-existence of dual anxiety and depression in difficult asthma is associated with poor consequences, both physically and psychologically. These findings highlight the urgent need for further interventional studies, both pharmacological and non-pharmacological, to effectively guide clinical practice in addressing the burden of anxiety and depression within difficult asthma [19].

Difficult asthma represents a complex constellation of comorbidities alongside asthma [10,11]. The multimorbidity implicated in difficult asthma is amply reflected in the WATCH cohort [20]. Such comorbidities appear to show particular patterns of association with clinical asthma phenotypes. For example, in WATCH, late-onset/female difficult asthmatics had a higher prevalence of obesity and poor disease-related questionnaire scores [20]. By phenotyping our difficult asthma cohort using anxiety and depression, we found that those phenotypes were associated with obesity, OSA, dysfunctional breathing and smoking, particularly in the dual anxiety and depression group (Figure 1). Many of these comorbidities are interrelated. Obesity is thought to have a bidirectional relationship with depression [21], with reports of obesity predicting the development of depression and vice versa [21]. Similarly, obesity is also correlated with OSA [22] and smoking [23]. Furthermore, smoking has also been shown to be linked to depression in a bidirectional manner, whereby depression may act as a vulnerability factor for smoking initiation and smoking as a vulnerability factor for developing depression [24]. An association between anxiety and dysfunctional breathing in asthma is known and is further supported by our observations here and elsewhere [5,25]. While the exact causality between these physical comorbidities, difficult asthma and anxiety and depression are difficult to ascertain, this complex web of interconnectivity highlights the need for a multidisciplinary approach targeting treatable traits [10,26,27], which would align with recommendations from the Global Initiative for Asthma difficult asthma guidelines [16]. Indeed, Burke et al., using a multidisciplinary (consultant, specialist nurse, physiotherapist and psychologist) approach, were able to reduce the number of hospital admissions in a subgroup of difficult asthma patients with psychological comorbidity [28]. This was also similarly demonstrated by Sastre et al., whereby specialist asthma input improved both asthma outcomes and the patient’s psychological disease state [29].

The anxiety alone and depression alone groups were both associated with worse disease-related questionnaire scores. While the association between anxiety and depression with poorer questionnaire scores has been well described in asthma [4,12,30], our findings extend it to difficult asthma, where the evidence is sparse. However, worse questionnaire scores were the only commonality shared between the two groups. Compared to controls, dysfunctional breathing was associated with anxiety alone but not depression alone. The greater prevalence of dysfunctional breathing may partly explain the higher number of hospitalisations observed. Dysfunctional breathing is associated with increased healthcare utilisation, worse quality of life and worse asthma control [10,25]. Additionally, both anxiety and dysfunctional breathing are associated with increased symptom perception, which recent cognitive models suggest may lead to suboptimal disease management [31,32]. Patients with depression alone, but not anxiety alone, had significantly worse lung function. There are several potential hypotheses for this. Depression has been linked with poor medication adherence [33]. Alternatively, the increased upregulation of Th2 cytokines associated with depressive states may have contributed to the observed worse lung function [34]. Furthermore, we found that the depression group had a higher prevalence of OSA and obesity, which are associated with systemic inflammation and irreversible airway obstruction [10,35]. However, this disparity is more likely due to the confounding effect of the greater prevalence of smoking in these patients, which was somewhat confirmed by the paradoxically lower FeNO observed, as smoking is known to reduce nitric oxide production [36]. Additionally, a lower prevalence of nasal polyposis was observed in the depression group but not the anxiety alone group. This might be associated with the higher observed prevalence of obesity, which has been shown by data from the British Thoracic Society Difficult Asthma Registry [37] to have an inverse association with nasal polyposis. To our knowledge, we are the first to describe the differential associations of anxiety in isolation and depression in isolation in difficult asthma. Although the exact mechanisms behind these differential associations are not fully understood, our data suggest that an individualised approach is warranted to address the disease-specific needs [26]. For example, we have previously shown that psychological interventions such as cognitive—behavioural therapy and mindfulness could be well integrated with pharmacotherapy to improve difficult asthma outcomes but needed to be appropriately tailored to meet disease-specific needs [38,39].

The dual anxiety and depression group is particularly noteworthy. They represented the majority (54.7%, 104/190) of our patients with anxiety or depression, which is unsurprising, given that both diseases are well known to co-exist [40]. Studies have shown that ≥70% of patients with depression have anxiety symptoms, while 49% of those with an anxiety disorder also had comorbid depression [40]. Additionally, only the dual disease group was associated with a female sex predominance, early age-of-onset and being younger in age. These findings mirrored our recent characterisation of the WATCH cohort using age-of-onset/sex, whereby the early-onset/female group had the highest prevalence of psychological comorbidities [20]. An oversimplified explanation of the female sex predominance observed may be the potential complex interactions of biological factors encompassing sex hormones, chronic inflammation and psychosocial factors including healthcare-seeking behaviours and symptom perception [20,41]. Furthermore, worse HADS-D scores in this group were associated with additional, unique features of poor asthma outcomes: higher prevalence of ICU admissions, intubation and mOCS dependency. Furthermore, only this group showed significantly worse SGRQ symptom scores and significant associations with additional (non-depression/anxiety) psychological comorbidities. They also had the highest proportion of patients within the severe HADS-A and HADS-D strata. Together, this highlights that the co-existence of both anxiety and depression has poor implications in difficult asthma, both physically and psychologically. Indeed, most other studies investigating the impact of psychological comorbidities in asthma of varying severity have focussed on anxiety and depression in combination, which is associated with poorer outcomes [4,5,8,9,12,30]. Additionally, our study of biologics use in WATCH found that these psychological comorbidities were associated with a poor biologics response [42]. Furthermore, our findings associating dual disease with worse psychological outcomes are supported by studies in mental health and primary care settings [43,44]. The poor outcomes seen in this group may be due to a possible compounding effect between anxiety and depression. Alternatively, this could also arise from, or be aggravated by, the multimorbidity associated with this group. The high prevalence of smoking, obesity, OSA, dysfunctional breathing and the complex interplay between these comorbidities may have contributed to the observed worse asthma morbidity. However, instead of being the driving force behind worse outcomes, dual anxiety and depression may instead be the unfortunate sequelae of chronic, poorly controlled asthma. Individuals with chronic diseases are more likely to develop psychological comorbidities [45]. Indeed, an important qualitative study detailing the life experiences of severe asthma patients described the significant emotional distress they experience [46]. The significantly younger age-of-onset in the dual-disease group suggests that the associations between anxiety and depression could be a consequence of the long-suffering disease. This is supported by the independent association between dual disease with longer asthma duration in multivariate analysis. Regardless of causality, our findings point to an intricate, vicious cycle connecting difficult asthma and physical comorbidities with dual anxiety and depression that worsens the burden of each condition alone, leading to an abysmal quality of life [19]. In light of this, difficult asthma physicians should screen patients with depression for anxiety and vice versa to guide disease-specific personalised therapies [10,26].

A novel approach of this study was to stratify patients by HADS-A and HADS-D scores. Findings from this approach suggest a possible severity-graded relationship between HADS-D, HADS-A and difficult asthma outcomes. Ciprandi et al. found that, in a general asthma outpatient setting, asthma control test scores were significantly inversely correlated with both HADS-A and HADS-D scores [47]. Our data echo these findings, but additionally showed that quantitatively worse anxiety and depression were associated with worse outcomes specifically in difficult asthma. The impact of psychological comorbidity severity on difficult asthma outcomes was further emphasised when we compared those with either ‘severe’ anxiety or depression against those with no anxiety and/or depression. This similarly showed that the most severe disease was associated with worse difficult asthma outcomes and independently associated with worse asthma control, strengthening our aforementioned observations. Additionally, our data highlight that the severity-gradient relationship between anxiety and depression and asthma diverges according to psychological disease. Worse HADS-D scores in those with depression or dual anxiety and depression were associated with worse subjective and objective difficult asthma outcomes. Conversely, in those with anxiety or dual anxiety and depression, worse HADS-A scores were only associated with worse subjective difficult asthma outcomes. This divergence again emphasises the differential effects of anxiety and depression on difficult asthma outcomes. It may also suggest that depression has a more potent biological impact on asthma than does anxiety, although work investigating innate cytokine production capacity in depressive and anxiety disorders found no difference between the two [48]. Our findings highlight the utility of validated psychological tools in the characterisation of difficult asthma patients, which may help clinicians identify which patients may require specialist psychological services as part of their asthma management. These tools may also be useful to screen difficult asthma patients for missed anxiety and depression. Indeed, our data showed that 9.4% of our controls had potentially undiagnosed moderate-to-severe depression and 5.1% had potentially undiagnosed moderate-to-severe anxiety, when assessed by HADS scores. Uncovering such comorbidities, especially if severe, may directly help improve asthma control and clinical outcomes. An example of this was a recent randomised controlled trial by Brown et al., whereby severe asthma patients with co-morbid major depression who were treated with 12-week continuous escitalopram therapy not only had reductions in depressive symptomology but also had significant improvements in asthma outcomes compared to the placebo arm [49]. This again emphasises the clinical importance of upfront characterisation of psychological comorbidity, followed by personalised therapy to address both co-existent pathologies [19,49,50,51].

Our study had limitations. We did not have qualitative nor detailed psychopharmacology data on our patients, which may have provided additional insights. We were also limited by the existence of missing data. However, this limitation is inherent in real-world observational studies and reflects clinical practice. Another limitation was that the characterisation was based on chart review and historical, clinician-based diagnosis of psychological comorbidity, which may harbour inaccuracies. Indeed, this may be partly the reason some patients with mild-to-moderate HADS scores did not have a diagnostic label of psychological disease. However, while imperfect, this again is reflective of real-life clinical practice. Additionally, our assessments were cross-sectional, at the enrolment time point, so we did not have data on the time-order relationship between the psychological comorbidities and difficult asthma. This may partly explain the ‘normal’ HADS scores of some of our patients with anxiety and depression, as their psychological comorbidity may have been well controlled at assessment. Future longitudinal work is needed to clarify such associations. Another limitation is that the association between anxiety and depression and difficult asthma outcomes may likely be somewhat confounded by physical comorbidities such as obesity. However, this is a reflection of real-life difficult asthma which is a complex web of interconnectivity that warrants a holistic, multidisciplinary approach. Our study had several strengths. We are the first to thoroughly phenotype difficult asthma patients using the presence of anxiety and depression and to undertake detailed stratification of those patients using HADS-A and HADS-D scores. Furthermore, the WATCH cohort is a large, real-world difficult asthma cohort and thereby the findings have high clinical translatability.

In conclusion, our findings demonstrate that anxiety and depression have a negative association with difficult asthma outcomes. Additionally, we showed that anxiety alone, depression alone and dual disease exhibited differential associations with difficult asthma. Furthermore, we found that dual anxiety and depression in the context of difficult asthma was associated with multimorbidity, worse asthma outcomes and worse psychological outcomes. We also demonstrated a severity-gradient relationship between anxiety and depression and difficult asthma outcomes. Collectively, our findings provide a thorough real-world perspective on the effect of anxiety and depression on difficult asthma patients and highlight the need for a targeted, holistic, multidisciplinary approach, combining both physical and psychological interventions and expertise in tackling such comorbidities. Future work should focus on formulating practical and effective treatment strategies to address these treatable traits. Work is also needed to clarify the directions of association between anxiety and depression, difficult asthma and associated comorbidities. Meanwhile, difficult asthma patients with dual anxiety and depression should be prioritised for comprehensive adjunctive therapy for their anxiety and depression, given their associated multimorbidity and poor outcomes.

## Figures and Tables

**Figure 1 jpm-12-00686-f001:**
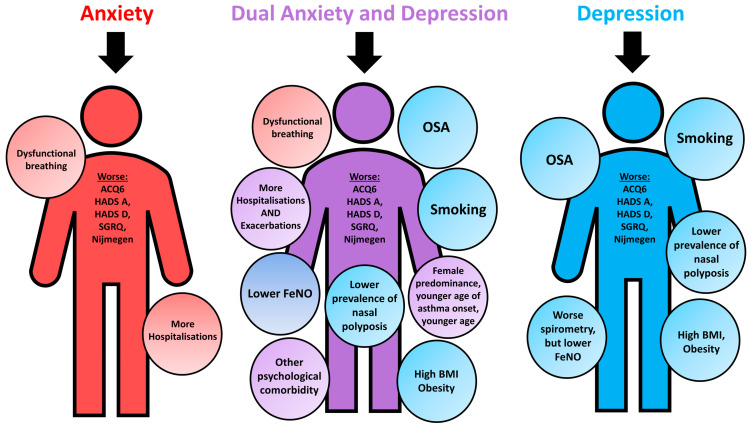
Summary clinical and phenotypical characteristics of WATCH subjects with psychological comorbidities, characterised against controls. ACQ6, Asthma Control Questionnaire 6; BMI, body mass index; HADS-A, Hospital Anxiety and Depression Scale Anxiety score; HADS-D, Hospital Anxiety and Depression Scale Depression score; SGRQ, St. George’s Respiratory Questionnaire; FeNO; fractional exhaled nitric oxide; OSA, obstructive sleep apnoea.

**Figure 2 jpm-12-00686-f002:**
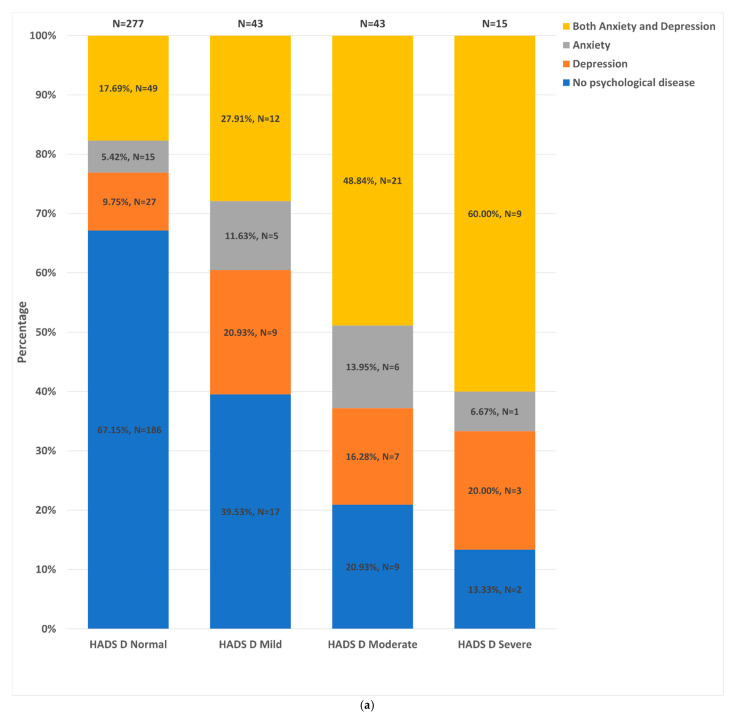
Distribution of the different patient groups across the HADS-D and HADS-A strata. (**a**). Distribution of HADS-D strata: HADS normal = HADS score of 0–7; HADS mild = HADS score of 8–10; HADS moderate = HADS score of 11–14; HADS severe = HADS score ≥15. (**b**). Distribution of HADS-D strata: HADS-D normal = HADS score of 0–7; HADS mild = HADS score of 8–10; HADS moderate = HADS score of 11–14; HADS severe = HADS score ≥15.

**Table 1 jpm-12-00686-t001:** Phenotypic characteristics of WATCH patients by clinical diagnoses of psychological comorbidity.

	Anxiety Alone (*N* = 35, 6.99%)		Depression Alone (*N* = 51, 10.7%)		Dual Anxiety and Depression (*N* = 104, 20.8%)		Control (*N* = 251, 50.1%)	
	Median (IQR)	Missing	Median (IQR)	Missing	Median (IQR)	Missing	Median (IQR)	Missing
Age of asthma onset—y	28.5 (37.0)	3	20.5 (37.0)	1	11.0 ** (24.0)	5	24.0 (39.0)	11
Duration of asthma at enrolment—y	23 (19.5)	3	25 (34.75)	1	26.0 (17)	5	24.0 (29)	11
Current age—y	57.0 (17.0)	0	52.0 (20.0)	0	48.0 ** (22.0)	0	58.0 (22.0)	0
Body mass index—kgm^−2^	30.6 (6.50)	0	32.4 ** (13.5)	0	31.1 ** (10.0)	3	28.3 (8.50)	2
	Percentage (frequency)	Missing	Percentage (frequency)	Missing	Percentage (frequency)	Missing	Percentage (frequency)	Missing
Sex, male	34.3% (12)	0	29.4% (15)	0	21.2% ** (22)	0	41.8% (105)	0
Obesity (body mass index ≥ 30), ever	54.3% (19)	0	70.6% ** (36)	0	60.4% ** (61)	3	41.4% (103)	2
Smoked, ever	54.3% (19)	0	64.7% ** (33)	0	53.8% ** (56)	0	39.6% (99)	1
Rhinitis, ever	64.7% (22)	1	54.3% (25)	5	66.0% (66)	4	67.6% (161)	13
GORD, ever	71.4% (25)	0	56.0% (28)	1	73.1% (76)	0	59.4% (148)	2
Eczema, ever	25.7% (9)	0	21.6% (11)	0	27.9% (29)	0	26.9% (67)	2
Urticaria/angioedema, ever	5.88% (2)	1	3.92% (2)	0	12.5% (13)	0	8.80% (22)	1
Dysfunctional breathing, ever	61.8% * (21)	1	45.1% (23)	0	65.7% ** (67)	2	37.0% (90)	8
Intermittent laryngeal obstruction, ever	16.1% (5)	4	12.0% (6)	1	20.4% (20)	6	11.6% (27)	18
Obstructive sleep apnoea, ever	5.88% (2)	1	11.8% ** (6)	0	13.5% ** (14)	0	4.03% (10)	3
COPD, ever	17.6% (6)	1	7.84% (4)	0	9.62% (10)	0	9.60% (24)	1
Bronchiectasis, ever	9.09% (3)	2	11.8% (6)	0	9.62% (10)	0	16.7% (42)	0
Positive for aeroallergen SPT, ever	61.3% (19)	4	68.6% (24)	16	73.8% (59)	24	67.2% (131)	56
Sulphite sensitivity, ever	2.94% (1)	1	3.92% (2)	0	10.7% (11)	1	8.00% (20)	1
Salicylates sensitivity, ever	20.6% (7)	1	21.6% (11)	0	30.8% (32)	0	24.0% (60)	1
Nasal polyposis, ever	16.1% (5)	4	8% (4) **	1	16.2% (16) *	5	30.2% (71)	16

Significance when compared to the control group: * *p* < 0.050; ** *p* < 0.005. Continuous variables are expressed as medians and interquartile ranges (IQR). Categorical variables are expressed as percentages (frequency). GORD, gastro—oesophageal reflux disease; COPD, chronic obstructive pulmonary disease; SPT, skin prick test. Continuous variables were analysed with Mann—Whitney U tests. Categorical variables were analysed by chi-squared test or Fisher’s exact test.

**Table 2 jpm-12-00686-t002:** Clinical measures of WATCH patients by clinical diagnoses of psychological comorbidity.

	Anxiety Alone (*N* = 35, 6.99%)		Depression Alone (*N* = 51, 10.7%)		Dual Anxiety and Depression (*N* = 104, 20.8%)		Control (*N* = 251, 50.1%)	
	Mean (SD)/ Median (IQR)	Missing	Mean (SD)/ Median (IQR)	Missing	Mean (SD)/ Median (IQR)	Missing	Mean (SD)/ Median (IQR)	Missing
ACQ6 at enrolment, median (IQR)	3.00 * (2.20)	4	3.20 * (1.30)	2	2.80 ** (1.60)	9	2.20 (1.80)	14
HADS-A at enrolment, median (IQR)	10.0 ** (5.00)	9	7.00 * (5.00)	5	11.0 ** (8.00)	14	4.00 (6.00)	38
HADS-D at enrolment, median (IQR)	5.00 * (9.00)	8	6.50 ** (7.00)	5	7.00 ** (8.00)	13	4.00 (5.00)	37
SGRQ score (Symptoms) at enrolment, mean (SD)	68.60 (24.05)	3	67.07 (20.52)	9	70.69 ** (18.32)	20	60.23 (23.65)	44
SGRQ score (Activity) at enrolment, mean (SD)	70.03 * (28)	6	74.93 ** (19.05)	10	72.28 ** (21.94)	23	55.48 (25.85)	60
SGRQ score (Impacts) at enrolment, mean (SD)	44.96 * (23.68)	5	44.95 ** (19.59)	10	50.10 ** (20.02)	22	34.76 (20.73)	54
SGRQ score (Total) at enrolment, mean (SD)	57.12 ** (22.92)	6	57.28 ** (17.34)	11	60.29 ** (17.81)	24	45.35 (20.05)	65
Nijmegen score at enrolment, median (IQR)	28.0 ** (16.0)	9	22.0 * (18.0)	14	30.0 ** (17.0)	26	17.0 (15.0)	64

Significance when compared to the control group: * *p* < 0.050; ** *p* < 0.005. Continuous variables are expressed as medians and interquartile ranges (IQR) or means and standard deviations (SD). Categorical variables are expressed as percentages (frequency). ACQ6, Asthma Control Questionnaire 6; HADS-A, Hospital Anxiety and Depression Scale Anxiety score; HADS-D, Hospital Anxiety and Depression Scale Depression score; SGRQ, St. George’s Respiratory Questionnaire. Continuous variables were analysed with Mann—Whitney U or independent *t*-tests where appropriate. Categorical variables were analysed by chi-squared test or Fisher’s exact test.

**Table 3 jpm-12-00686-t003:** Asthma healthcare utilisation and outcomes of WATCH patients by clinical diagnoses of psychological comorbidity.

	Anxiety Alone (*N* = 35, 6.99%)		Depression Alone (*N* = 51, 10.7%)		Dual Anxiety and Depression (*N* = 104, 20.8%)		Control (*N* = 251, 50.1%)	
	Median (IQR)	Missing	Median (IQR)	Missing	Median (IQR)	Missing	Median (IQR)	Missing
Number of asthma exacerbations in the prior 12 months	4.00 (5.00)	3	4.00 (5.00)	4	3.00 * (5.00)	8	3.00 (3.00)	25
Number of asthma hospitalisations in the prior 12 months	0.00 * (2.00)	1	0.0 (1.00)	1	0.00 * (1.00)	0	0.00 (0.00)	1
Number of days taken off work/education in the prior 12 months due to asthma	7.00 (20.0)	14	13.0 (61.0)	30	7.00 (30.0)	51	4.00 (14.0)	108
	Percentage (frequency)	Missing	Percentage (frequency)	Missing	Percentage (frequency)	Missing	Percentage (frequency)	Missing
On maintenance OCS at enrolment, yes	31.3% (10)	3	32.0% (16)	1	30.9% (30)	7	32.0% (77)	10
ICU admissions for asthma, ever	28.6% (10)	0	23.5% (12)	0	29.8% (31)	0	25.1% (63)	0
Intubation for asthma, ever	17.1% (6)	0	12.0% (6)	1	11.5% (12)	0	12.4% (31)	1
On asthma biologics at enrolment, yes	8.6% (3)	0	7.8% (4)	0	15.4% (16)	0	19.1% (48)	0

Significance when compared to the control group: * *p* < 0.050. Continuous variables expressed as medians and interquartile ranges (IQR). Categorical variables expressed as percentages (frequency). OCS, oral corticosteroids; ICU, intensive care unit. Continuous variables were analysed with Mann—Whitney *U* tests. Categorical variables were analysed by chi-squared test or Fisher’s exact test.

**Table 4 jpm-12-00686-t004:** Factors independently associated with anxiety and depression and severe psychological comorbidities.

Model Name	Cases Included	Variables Included	Final Variables	*p*-Value	OR; 95 CI
Dual anxiety and depression vs. controls	230/355 64.8%	Sex; Age; Age at asthma diagnosis; Smoking, ever; OSA; Nasal polyposis; Dysfunctional breathing; BMI; Duration of asthma at enrolment; Other psychiatric comorbidities; ACQ6; Exacerbations; Hospitalisations	Sex	0.001	0.23 (0.09−0.56)
			Age	0.004	0.97 (0.94−0.99)
			Smoking, ever	0.011	2.48 (1.23−5.00)
			OSA	0.021	4.08 (1.23−13.46)
			Duration of asthma at enrolment	0.023	1.03 (1.00−1.05)
			Nasal polyposis	0.058	0.42 (0.17−1.03)
			Other psychiatric comorbidities	0.038	3.17 (1.07−9.41)
			ACQ6	0.068	1.28 (0.98−1.67)
Severe anxiety and/or depression vs. no anxiety and/or depression	118/185 63.8%	Sex; Age; Age at asthma diagnosis; Smoking, ever; Brochiectasis; Nasal polyposis; Eczema; OSA; Dysfunctional breathing; BMI GORD; Other psychiatric comorbidities; Exacerbations; ACQ6	Sex	0.019	0.13 (0.02−0.71)
			Smoking, ever	0.067	3.54 (0.91−13.68)
			ACQ6	<0.001	3.66 (1.99−6.73)

BMI, body mass index; Exacerbations, exacerbations requiring OCS/increase in maintenance OCS in the prior 12 months from enrolment; ACQ6, Asthma Control Questionnaire 6; GORD, gastro-oesophageal reflux disease; OSA, obstructive sleep apnoea; other psychological comorbidities, psychological comorbidities which are not anxiety/depression; ‘severe’ anxiety and/or depression, defined as a clinical diagnosis of anxiety or depression and ‘severe’ HADS-A or HADS-D scores; No anxiety and/or depression, defined as no anxiety, no depression and ‘normal’ HADS-A and HADS-D scores. Multiple logistic regression was performed (backward stepwise variable selection) using variables trending towards significance (*p* < 0.1).

**Table 5 jpm-12-00686-t005:** Asthma outcomes of those with a clinical diagnosis of depression by HADS-D stratification.

	HADS-D Normal (0–7) *N* = 79, 54.1%		HADS-D Mild (8–10) *N* = 25, 17.1%		HADS-D Moderate (11–14) *N* = 30, 20.5%		HADS-D Severe (≥15) *N* = 12, 8.2%		*p*-Value
	Mean (SD)/ Median (IQR)	Missing	Mean (SD)/ Median (IQR)	Missing	Mean (SD)/ Median (IQR)	Missing	Mean (SD)/ Median (IQR)	Missing	
Number of exacerbations in the prior 12 months, median (IQR)	2.5 (5)	5	3 (4)	2	4 (6)	2	4.5 (4.25)	2	0.099
Number of days lost from work/education in the prior 12 months, median (IQR)	5 (28)	34	14 (80)	16	7.5 (45)	16	29 (NA)	10	0.464
Number of hospitalisations the prior 12 months, median (IQR)	0 (1)	0	0 (1)	1	0 (1)	0	1 (2)	0	0.293
ACQ6 at enrolment, median (IQR)	*2.7* (1.8)	1	3.4 (1.3)	1	3.2 (1.45)	1	**4.7** (1.4)	1	<0.001
SGRQ score (Symptoms) at enrolment, mean (SD)	*64.51* (18.60)	9	73.11 (13.81)	3	72.63 (21.67)	1	**79.69** (11.86)	4	0.031
SGRQ score (Activity) at enrolment, mean (SD)	*64.43* (22.91)	11	83.84 (13.70)	4	78.58 (13.46)	3	**92.82** (18.03)	4	<0.001
SGRQ score (Impacts) at enrolment, mean (SD)	*39.78* (17.52)	10	53.72 (17.45)	3	54.65 (17.08)	3	**73.36** (18.65)	4	<0.001
SGRQ score (Total) at enrolment, mean (SD)	*51.02* (16.20)	12	66.43 (13.85)	4	64.65 (13.95)	3	**80.43** (14.47)	4	<0.001
	Percentage (frequency)	Missing	Percentage (frequency)	Missing	Percentage (frequency)	Missing	Percentage (frequency)	Missing	*p*-value
ICU admissions for asthma, ever	21.5% (17)	0	*20%* (5)	0	**60%** (18)	0	25% (3)	0	0.001
Intubation for asthma, ever	7.6% (6)	0	4% (1)	0	23.3% (7)	0	16.7% (2)	0	0.064
On maintenance OCS at enrolment, yes	27.4% (20)	6	*12.5%* (3)	1	46.7% (14)	0	**81.8%** (9)	1	<0.001

Continuous variables are expressed in medians and interquartile ranges (IQR) or means and standard deviations (SD) and were analysed with one-way ANOVA or Kruskal—Wallis test and post hoc analyses. Categorical variables expressed as percentages (frequency) and analysed by chi-squared tests and post hoc analyses. For significant differences, highest values indicated in bold and lowest values in italics. OCS, oral corticosteroids; ACQ6, Asthma Control Questionnaire 6; ICU, intensive care unit.

**Table 6 jpm-12-00686-t006:** Asthma outcomes for those with a clinical diagnosis of dual anxiety and depression by HADS-D stratification.

	HADS-D Normal (0–7) *N* = 49, 53.8%		HADS-D Mild (8–10) *N* = 12, 13.2%		HADS-D Moderate (11–14) *N* = 21, 23.1%		HADS-D Severe (≥15) *N* = 9, 9.9%		*p*-Value
	Mean (SD)/Median (IQR)	Missing	Mean (SD)/Median (IQR)	Missing	Mean (SD) Median (IQR)	Missing	Mean (SD)/Median (IQR)	Missing	
Number of exacerbations in the prior 12 months, Median (IQR)	*2* (4)	4	*2* (4)	1	5 (6.5)	0	4 (4.5)	1	0.016
Number of days lost from work/education in the prior 12 months, Median (IQR)	6 (28.5)	19	3.5 (138.25)	8	5 (30)	10	29 (NA)	7	0.596
Number of hospitalisations the prior 12 months, median (IQR)	0 (1)	0	0.5 (1)	0	0 (1)	0	1 (1.5)	0	0.626
ACQ6 at enrolment, median (IQR)	*2.7* (1.75)	1	2.9 (1.3)	0	3.25 (1.3)	1	**4.5** (1.43)	1	0.001
SGRQ score (Symptoms) at enrolment, mean (SD)	*66.02* (18.61)	5	70.81 (11.46)	0	78.43 (15.70)	1	**84.54** (10.30)	4	0.015
SGRQ score (Activity) at enrolment, mean (SD)	*65.44* (22.14)	5	78.66 (13.84)	1	82.91 (13.32)	3	**89.70** (23.03)	4	0.002
SGRQ score (Impacts) at enrolment, mean (SD)	*41.72* (17.23)	5	52.09 (18.02)	0	60.59 (12.65)	3	**80.28** (16.49)	4	<0.001
SGRQ score (Total) at enrolment, mean (SD)	*52.92* (15.68)	6	63.79 (14.03)	1	70.06 (10.27)	3	**83.87** (16.16)	4	<0.001
	Percentage (frequency)	Missing	Percentage (frequency)	Missing	Percentage (frequency)	Missing	Percentage (frequency)	Missing	*p*-value
ICU admissions for asthma, ever	20.4% (10)	0	*16.7%* (2)	0	**57.1%** (12)	0	33.3% (3)	0	0.014
Intubation for asthma, ever	2% (1)	0	*0%* (0)	0	**28.6%** (6)	6	22.2% (2)	0	0.002
On maintenance OCS at enrolment, yes	25% (11)	5	*9.1%* (1)	1	47.6% (10)	0	**87.5%** (7)	1	0.001

Continuous variables are expressed as medians and interquartile ranges (IQR) or means and standard deviations (SD) and analysed with one-way ANOVA or Kruskal—Wallis test and post hoc analyses. Categorical variables expressed as percentages (frequency) and analysed by chi-squared tests and post hoc analyses. For significant differences, highest values indicated in bold and lowest values in italics. OCS, oral corticosteroids; ACQ6, Asthma Control Questionnaire 6; ICU, intensive care unit.

## Data Availability

Data are available upon reasonable request from the corresponding author.

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
