# Peer review of "The Detrimental Clinical Associations of Anxiety and Depression with Difficult Asthma Outcomes"

_jpm, 2022, doi:10.3390/jpm12050686_

Round 1
Reviewer 1 Report
ABSTRACT
Well-described and informative abstract.
- Change “introduction” to objective” in line 22
- Define what do you mean by difficult asthma outcomes in line 22
- Define what is difficult asthma, any severity scale or specific diagnostic codes were used?
INTRODUCTION
Needs more work. Add more background info as asked below followed by continuity of current content.
- The initial content should focus on the epidemiology of asthma and the prevalence of psychiatric conditions with more details on anxiety and depression in this population.
- More info from biological studies on underlying pathophysiology link between anxiety and depression with asthma.
METHODS
Well-described criteria for difficult asthma, and recommend to add a brief version of this definition in abstract as well. Recommend to split the current content into study sample, setting, inclusion and exclusion criteria, primary outcome, secondary outcomes, analysis (already there), and ethical approval. The analysis and data variables have been well described and robust statistical models have been used.
RESULTS
The content needs to elaborate more from the tables, and also a rearrangement of the structure is recommended as can be followed through my comments below.
- The first paragraph should be of the total sample characteristics and recommend creating a table of only demographics (dual and control).
- Table 1: demographic portion can be separated as table 1 and the outcome portion as Table 2. Also, missing numbers (pls remove as a column) as it can be calculated as proportion % per variable and mentioned in a footnote.
- Many variables that may be important are not described as content from table 1.
- Extract some statistical points from the impact of depression and anxiety on the outcome scales used in Table 2 and describe them in layman terms. Authors have only mentioned these for anxiety alone, but pls also describe these outcomes for depression too.
- Recommend to start with dual anxiety and depression vs controls as starting point and change table 4 to table 1.
- Too many tables. Recommend to change at least 1-2 tables that are less important and if do not serve as main goals of study to be in supplemental data. Information provided by authors is respected but will lose the interest of readers
DISCUSSION
Good work was done by the authors to explain their result findings and give scientific reasoning based on existing literature. Limitations have been well explained, and the conclusion is pertaining strictly to the study results. I recommend adding some clinical implications about collaborative care practice between allergy/medicine and psychiatry to improve patient outcomes.
Author Response
ABSTRACT
Comment: Well-described and informative abstract.
- Change “introduction” to objective” in line 22
- Define what do you mean by difficult asthma outcomes in line 22
- Define what is difficult asthma, any severity scale or specific diagnostic codes were used?
Response: We thank the reviewer for the positive comment. We have made the suggested changes. We have also removed subheadings as suggested by Reviewer 2. Please see the ‘abstract’ section for more details.
INTRODUCTION
Comment: Needs more work. Add more background info as asked below followed by continuity of current content.
- The initial content should focus on the epidemiology of asthma and the prevalence of psychiatric conditions with more details on anxiety and depression in this population.
- More info from biological studies on underlying pathophysiology link between anxiety and depression with asthma.
Response: We thank the reviewer for this suggestion and have amended the Introduction to accommodate these helpful suggestions. Please see lines 54-61 of the Introduction. We would highlight that understanding of pathophysiological links of anxiety and depression with asthma is limited and in it’s infancy.
METHODS
Comment: Well-described criteria for difficult asthma, and recommend to add a brief version of this definition in abstract as well. Recommend to split the current content into study sample, setting, inclusion and exclusion criteria, primary outcome, secondary outcomes, analysis (already there), and ethical approval. The analysis and data variables have been well described and robust statistical models have been used.
Response: We thank the reviewer for their positive review and comment on our methods. While we acknowledge the reviewer’s point, we respectfully think that such a layout would be more suited for report on a randomized controlled trial. In our humble opinion such a layout is less suitable for this descriptive report and have therefore not adopted the suggestion.
RESULTS
Comment: The content needs to elaborate more from the tables, and also a rearrangement of the structure is recommended as can be followed through my comments below.
- The first paragraph should be of the total sample characteristics and recommend creating a table of only demographics (dual and control).
- Table 1: demographic portion can be separated as table 1 and the outcome portion as Table 2. Also, missing numbers (pls remove as a column) as it can be calculated as proportion % per variable and mentioned in a footnote.
Response: We thank the reviewer for these suggestions. We have added overall sample characteristics into the first paragraph as advised. We note request to remove the missing column as it would be possible to impute that information from the proportions listed. We would highlight that in a real-world study, that the sample size may vary from parameter to parameter as participants may not have complete data for all variables. In other words the denominator may not be fixed and thus imputation of missingness from the proportions may not be feasible. It is also not possible to impute missingness for the continuous variables listed in the table. Hence we have not removed the missing columns. Furthermore, respectfully we have not separated Table 1 as this would generate more tables which would contradict the reviewer’s comment 6 to reduce the number of Tables. We also think that Table 1 being a presentation of phenotypical and demographic data is appropriately housed in a single table.
Comment: Many variables that may be important are not described as content from table 1.
Response: We thank the reviewer for this comment. It is unclear what else exactly they refer to. We have presented what data is available from our cohort study in the table. Please see our additions to lines 144-148, 189-192, 207-209 to further clarify both positive and negative findings in the respective tables.
Comment: Extract some statistical points from the impact of depression and anxiety on the outcome scales used in Table 2 and describe them in layman terms. Authors have only mentioned these for anxiety alone, but pls also describe these outcomes for depression too.
Response: We thank the reviewer for this suggestion. This can already be found in line 164-168 of the manuscript, between Tables 2 and 3 in the draft manuscript.
Comment: Recommend to start with dual anxiety and depression vs controls as starting point and change table 4 to table 1.
Response: We thank the reviewer for this suggestion. While we acknowledge the reviewer’s point, we respectively disagree with this approach. In our humble opinion, we favour the current way we have presented the data as this depicts a clear narrative whereby the individual conditions are described first in isolation and then in combination, to highlight how in combination having the two diseases are beyond cumulatively worse.
Comment: Too many tables. Recommend to change at least 1-2 tables that are less important and if do not serve as main goals of study to be in supplemental data. Information provided by authors is respected but will lose the interest of readers
Response: We thank the reviewer for this suggestion. It is not possible to easily reduce the number of tables but in response to concerns about information overload, we have moved some data from several tables into the supplementary data. In that regard, we have moved lung function, blood eosinophil, total IgE and FENO data into the supplementary material.
DISCUSSION
Comment: Good work was done by the authors to explain their result findings and give scientific reasoning based on existing literature. Limitations have been well explained, and the conclusion is pertaining strictly to the study results. I recommend adding some clinical implications about collaborative care practice between allergy/medicine and psychiatry to improve patient outcomes.
Response: We again thank the reviewer for their positive comments and their suggestion. We have alluded to the reviewer’s suggestion in lines 320-323 of the manuscript. We have also added a sentence on this in the conclusion section (lines 489-490).
Reviewer 2 Report
The topic of the manuscript is interesting and the approach innovative.
Minor points
Abstract
I suggest deleting the unnecessary subsessions (introduction, methods, etc..) that reduce the readability of the text.
Results
Table 1 can be improved. Why did the authors mix together continuous variables and categorical variables, maybe it could be better to split these measures into two distinct tables. In line 127 a sentence is repeated twice.
Why does the total number of patients differ in figures 2a and 2b with respect to previous reports (see table 1, table 2 and table and main manuscript text )?
Author Response
Comment: The topic of the manuscript is interesting and the approach innovative.
Response: We thank the reviewer for this positive comment.
Minor points
Abstract
Comment: I suggest deleting the unnecessary subsessions (introduction, methods, etc..) that reduce the readability of the text.
Response: We have done as suggested and now present the abstract as a single paragraph; please see the ‘abstract’ section of our manuscript.
Results
Comment: Table 1 can be improved. Why did the authors mix together continuous variables and categorical variables, maybe it could be better to split these measures into two distinct tables. In line 127 a sentence is repeated twice.
Response: We thank the reviewer for this suggestion. Respectfully, we have mixed both these variables as in our humble opinion, these are the demographic and phenotypic characteristics of our cohort which go well together. Splitting them into even more tables would in our opinion result in too many tables and would be fatiguing for the reader. The other reviewer has already suggested trying to reduce the number of tables and volume of data presented in the main manuscript.
Comment: Why does the total number of patients differ in figures 2a and 2b with respect to previous reports (see table 1, table 2 and table and main manuscript text )?
Response: We thank the reviewer for highlighting this so that we can offer some clarification. The numbers in Figure 2a and 2b are in reference to the number of patients who had HADS A and HADS D data available, while the numbers in Table 1, Table 2 represent the numbers without such limitations. To make this clearer we have added the following sentence ‘Of participants who had HADS-A and HADS-D data available, Figure 2a…..’ to that section of the results. Please see line 234.
Reviewer 3 Report
The article "The Detrimental Clinical Associations Of Anxiety and Depression with Difficult Asthma Outcomes" addresses a problem important from the perspective of public health, and the results of the study have application significance. They can facilitate early identification and management of anxiety and depression in difficult asthma patients.
The article is well written, I have only a few small comments below:
In lines 47-47 the authors write: "Despite clear associations between depression and anxiety with poor outcomes in asthma (1-6) ..." - The introduction requires at least some development. The authors write that the association between depression and asthma is clear, however, it should be precisely described. What means that it is clear? What are the results of the research done so far on this topic? A more detailed description of this issue is a better explanation for undertaking research on the relationship between depression and anxiety and difficult asthma.
In lines 73-76, the authors mention the names of the measurement tools but do not characterize them at all. I have read the article the authors refer to [reference 11], but the questionnaire methods are not described there either. All the more, I recommend supplementing this information. This will allow for a more in-depth understanding of the test results. At present, the complete knowledge of the names of measurement tools and their abbreviations is possible only by clinicians who use the above-mentioned measurement methods on a daily basis.
The manuscript also requires careful review in terms of text editing. For example, in at least some places in the body of the manuscript, extra spaces should be removed (i.e. lines: 238, 314, 326, 339, 343, 354, 369, 381, 385, 392, 393, 396, 407.
In line 393 there is also a lack of coma between "outcomes" and "An example ..."
Author Response
The article is well written, I have only a few small comments below:
Comment: In lines 47-47 the authors write: "Despite clear associations between depression and anxiety with poor outcomes in asthma (1-6) ..." - The introduction requires at least some development. The authors write that the association between depression and asthma is clear, however, it should be precisely described. What means that it is clear? What are the results of the research done so far on this topic? A more detailed description of this issue is a better explanation for undertaking research on the relationship between depression and anxiety and difficult asthma.
Response: We thank the reviewer for their positive comments about our paper and for raising this specific point. In order to address this we have added some further information in the Introduction, lines 54-64, to expand on some of the known associations of psychological comorbidity with asthma outcomes and to highlight the key knowledge gap in the context of difficult asthma.
Comment: In lines 73-76, the authors mention the names of the measurement tools but do not characterize them at all. I have read the article the authors refer to [reference 11], but the questionnaire methods are not described there either. All the more, I recommend supplementing this information. This will allow for a more in-depth understanding of the test results. At present, the complete knowledge of the names of measurement tools and their abbreviations is possible only by clinicians who use the above-mentioned measurement methods on a daily basis.
Response: We thank the reviewer for highlighting this helpful point and agree that it would considerably help deeper understanding and interpretation of the data. We have added some further interpretation to explain what these tests assess and how to interpret them (lines 90-94 in the revised manuscript).
Comment: The manuscript also requires careful review in terms of text editing. For example, in at least some places in the body of the manuscript, extra spaces should be removed (i.e. lines: 238, 314, 326, 339, 343, 354, 369, 381, 385, 392, 393, 396, 407.
Response: We thank the reviewer for noting this point. The extra spaces appear to be a function of the journal manuscript template and while we have tried to remove these, they have recurred in some places. We presume that these should be sorted when the final manuscript proofs undergo prepublication copy editing checks.
Comment: In line 393 there is also a lack of coma between "outcomes" and "An example ..."
Reply: We thank the reviewer for spotting this omission. We have inserted a full stop at the indicated point (line 452 in the revised manuscript).